# STCD-EffV2T Unet: Semi Transfer Learning EfficientNetV2 T-Unet Network for Urban/Land Cover Change Detection Using Sentinel-2 Satellite Images

**Masoomeh Gomroki** [1], **Mahdi Hasanlou** [1,*] and **Peter Reinartz** [2]

1  School of Surveying and Geospatial Engineering, College of Engineering, University of Tehran, Tehran 14174-66191, Iran
2  Deutsches Zentrum für Luft- und Raumfahrt (DLR), Institut für Methodik der Fernerkundung (IMF), 82234 Wessling, Germany
*  Correspondence: hasanlou@ut.ac.ir; Tel.: +98-21-6111-4525

**Abstract:** Change detection in urban areas can be helpful for urban resource management and smart city planning. The effects of human activities on the environment and ground have gained momentum over the past decades, causing remote sensing data sources analysis (such as satellite images) to become an option for swift change detection in the environment and urban areas. We proposed a semi-transfer learning method of EfficientNetV2 T-Unet (EffV2 T-Unet) that combines the effectiveness of composite scaled EfficientNetV2 T as the first path or encoder for feature extraction and convolutional layers of Unet as the second path or decoder for reconstructing the binary change map. In the encoder path, we use EfficientNetV2 T, which was trained by the ImageNet dataset. In this research, we employ two datasets to evaluate the performance of our proposed method for binary change detection. The first dataset is Sentinel-2 satellite images which were captured in 2017 and 2021 in urban areas of northern Iran. The second one is the Onera Satellite Change Detection dataset (OSCD). The performance of the proposed method is compared with YoloX-Unet families, ResNest-Unet families, and other well-known methods. The results demonstrated our proposed method's effectiveness compared to other methods. The final change map reached an overall accuracy of 97.66%.

**Keywords:** change detection; deep learning; EffIcientNetV2 T-Unet; semi transfer learning; Senteinel-2

## 1. Introduction

Remote sensing data are used in the detection of the changes unfolding in an area over a period of time [1]. This procedure, which is named change detection, has the capability to capture worthwhile information for environmental research, particularly research that is concentrated on urban development and smart cities, hydrological cycles, climate change, city management, and deforestation; therefore, it can be an essential factor for local and global development project plans [2]. In recent decades, human activities in the world have caused huge changes on the earth, causing remote sensing resources such as satellite images to become an important tool for simple, swift, and accurate change detection over vast areas. The methods which are used to gain information and detect changes through remote sensing data are categorized into two parts: semi-automatic and automatic [2].

The change detection techniques in images which are known as two-dimensional remote sensing data are generally grouped into two categories: supervised and unsupervised methods. The supervised methods are identified to be resistant to atmospheric condition changes, brightness, and weak sensor calibration during data capturing, which means that the supervised methods are not sensitive to radiometric correction but have a significant weakness when used on terrestrial reference data because collecting the reference data is time-consuming, costly, and complicated [2]. The unsupervised methods do not require

terrestrial reference data as they compare between pixels directly; however, they are sensitive to radiometric correction [2]. Table 1 briefly summarizes traditional change detection approaches and their advantages and limitations [3–6].

**Table 1.** Summary of the literature on traditional change detection approaches.

| Category | Subcategory | Definition | Mode | Advantages | Limitations | Applications |
|---|---|---|---|---|---|---|
| Visual Analysis | – | Generate a change map by visual interpretation | Supervised | Highly reliable results | Difficult to update, for large applications time-consuming, labor-intensive | Used in different fields before [3] |
| Algebra-based methods | Image differencing | A change map is generated by performing algebraic operations or transformations. | Unsupervised | Simple and easy to implement and interprets, it decreases the impact of sunglasses' topography shadow. | Difficult to choose the proper threshold to identify the change map, Difficult to choose appropriate image bands, there is no "from-to" change information, and the Distribution of results is not normal. | Urban land use [7], Urban land use and cover [8] |
| | Image regression | | | | | |
| | Image rationing | | | | | |
| | Chang vector analysis | | | | | |
| | Vegetation index differencing | | | | | |
| Transformation | Principle component analysis (PCA) | A change map is produced by using transformation methods; these methods are utilized to suppress correlated information and highlight variance. | Unsupervised | Reduce the redundancy between bands, emphasize different information in taken components | Detailed change information cannot extract | Rural-urban land cover [9], Land cover [10] |
| | Tasseled Cap | | | | | |
| | Chi-Square | | | | | |
| | Gramm-Schmidt | | | | | |
| Classification methods | Post-classification comparison | A change map is generated by a classification method | Supervised unsupervised hybrid | Provide change information matrix, do not need atmospheric correction | Selecting training data is challenging | Land cover [11], Urban land cover [12], Urban land cover [13], Forest change detection [14] |
| | Spectral-Temporal combined analysis | | | | | |
| | EM- transformed (Expectation Maximization) | | | | | |
| | Unsupervised change detection methods | | | | | |
| | Hybrid change detection | | | | | |
| | Artificial Neural Networks | | | | | |
| Advanced Method | Li-Strahler reflectance model | transform the spectral reflectance values into physically based parameters | hybrid | More straightforward to comprehend than the spectral signature. Can derive vegetation information | Complicated and time-consuming, developing the proper mode is challenging. | Land cover [3] |
| | Spectral mixture model | | | | | |
| | Biophysical parameter method | | | | | |
| GIS technique | Integrate GIS and RS methods | Use different data sources for change detection | hybrid | Land use information can update directly in the GIS database | The quality of the result change map depends on different data type | Forest change detection [15] |
| | GIS method | | | | | |

Given the merits and demerits of traditional change detection methods outlined in Table 1, it would be ideal to find a method which uses the features contained in images for change detection and thus improve on the drawbacks of these methods. In recent years, automatic deep learning networks have been highly effective in extracting high-level features from images for change detection purposes. Generally, deep learning methods for change detection include five categories: Convolutional Neural Networks (CNNs), Autoencoders (AEs) or stack Autoencoders (SAEs), Recurrent Neural Networks (RNNs), Generative Adversarial Networks (GANs), and Deep Belief Networks (DBNs) [16]. This study focuses on urban change detection of multispectral satellite images. The previous

research on deep learning change detection methods and their advantages and limitations of multispectral remote sensing datasets are investigated briefly in Table 2.

**Table 2.** Literature review of deep learning change detection approaches.

| Category | Sub-Category | Mode | Data | Advantages | Limitations | Applications |
|---|---|---|---|---|---|---|
| CNN | Deep brief network | Supervised | Multispectral SPOT-5 and Landsat images, google earth images | High accuracy | Time of processing | Urban land use and vegetation [17] |
| CNN | Fully convolutional Siamese network | Supervised | Multispectral Onera Sentinel-2 Satellite images (OSCD) | Trained end to end | Huge amount of training data | Urban land use [18] |
| CNN | Spectral, spatial joint learning network | Supervised | Multispectral Taizhou and Kunshan dataset | High performance | Huge amount of training data | Urban land use [19] |
| CNN | Siamese deep network with hybrid convolutional feature extraction module | Supervised | Multispectral ZY-3 and GF-2 satellite images | Extraction robust deep features | Could not separate pixel from its neighbor for classification | Urban rural-urban non-urban land use [20] |
| CNN | Bilinear convolutional neural network | Supervised | Multispectral Lansat-8 satellite images | End-to-end training | Generating label data is challenging | River and Waterland use [21] |
| CNN | Multidimensional CNN | Unsupervised | OSCD | End to end | Time-consuming | Urban land use land cover [22] |
| CNN | Feature difference convolutional neural network | Pre-trained | Multispectral Worldview-3, QuickBird and Ziyuan-3 satellite images | Powerful robustness and generalization ability | Require huge amount of pixel-level training samples | Urban land use [23] |
| CNN | Deep Siamese semantic segmentation network | Supervised | RGB building images | Decrease training sample issue | Poor performance in detecting the exact boundary of the building | Urban construction [24] |
| CNN | Semi-supervised Siamese network based on transfer learning | Pre-trained | Haiti earthquake QuickBird satellite images | Decrease computational cost | Error map | Urban land cover [25] |
| CNN | Attention mechanism-based deep supervision network | Supervised | Multispectral LEVIR CD dataset [26] | High performance | mode complex | Urban land cover [27] |
| CNN | Multi-Attention guided feature fusion network | Supervised | LEVIRCD [26], WHUCD [28] | Enhance deep feature extraction and fusion | A large number of parameters | Building change detection [29] |
| AE | Multispectral Unet | Supervised | OSCD | End to end | Low performance | Urban land use [4] |
| AE and RNN | Combination of Unet and robust Recurrent Networks such as LSTM | Supervised | OSCD | End to end | A large amount of training data | Urban land use [30] |
| AE | Unet | Supervised | Multispectral KompSAT-3 satellite images | Solve spectral distortion issue | Computational complexity | Urban land use and forest [31] |

**Table 2.** *Cont.*

| Category | Sub-Category | Mode | Data | Advantages | Limitations | Applications |
|---|---|---|---|---|---|---|
| AE | Deep self-attention fully efficient convolutional Siamese network | Supervised | Google Earth multispectral season varying dataset | End to end | Complexity of model | Urban land use [32] |
| AE | Developed from Unet and SeNet | Supervised | Multispectral Wuhan dataset from IKONOS | End to end | A large amount of training data | 2D and 3D building change detection [33] |
| AE | Intensely supervised attention-guided network | Supervised | LEVIRCD [26] | Extract deep features efficiently | Time-consuming | Building change detection [34] |
| AE | Hierarchical self-attention augmented the Laplacian pyramid | Supervised | Higres satellite images | Extracting deep features | Complex model | Urban land use [4] |
| AE | Feature regularized mask DeepLab | Supervised | LEVIRCD [26], GF-1 satellite images | End to end | Cannot detect edge properly | Building change detection [35] |
| AE | Boundary-aware Siamese network | Supervised | LEVIRCD [26] | End to end, sharp boundary | Complex model | Urban land use [36] |
| AE | Efficient Unet++ | Supervised | LEVIRCD [26], CD dataset [37] | Minimize computational parameters | Time-consuming | Urban land use [38] |
| AE | Multitask learning framework L-Unet | Supervised | OSCD | Solve the illumination differences problem and registration error | A poor performance, especially in preserving object shape | Urban change detection [39] |
| AE | Unet | Supervised | OSCD | Simple model, easy to implement | Cannot recognize small changes | Urban change detection [40] |
| RNN | Recurrent convolutional Neural Network | Supervised | Multispectral Taizhou dataset | End to end | Cannot extract all deep feature | Urban land use [41] |
| GAN | Generative discriminatory classified network | Supervised | Multispectral Worldview-2 and GF-1 satellite images | Decrease training sample issue | Complexity of model | Urban land use and water [42] |
| GAN | Deep GAN with improved DeepLabV3+ | Unsupervised | OSCD, Landsat-8, and google earth satellite images | high performance | Huge amount of training data | Urban land use [43] |
| GAN | Self-supervised conditional GAN | Semi-supervised | Multispectral Worldview-2 satellite images | Extract features at multiple resolutions | Model complexity | Urban land use [44] |
| GAN | Feature output space dual alignment | Supervised | LEVIRCD [26], WHUCD [28] | Address the problem of the pseudo changes | Super-parameters $\alpha$ and $\beta$ are issue | Building change detection [45] |
| DBN | Deep joint segmentation | Unsupervised | Multispectral Sentinel-2 and Pleiades images | Any labeled training pixel is not required | Time-consuming | Urban land use [46] |

Regarding Table 2, most studies have focused on AEs and CNNs for the change detection of multispectral remote sensing data. CNNs have some advantages for change detection purposes such as high accuracy, high performance, extract robust features, and end-to-end training; however, they have limitations such as requiring a time-consuming and huge amount of training data. One of the main networks in this category is the Siamese network. CNNs change detection networks are used for urban land cover or land use, building change detection, and water land use. AEs are simple models and employ end-to-end training which extracts deep features efficiently. AEs have some limitations such

as computational complexity, a requirement for a large amount of training data, and poor performance in boundary detection. Unet is one of the main AE networks for change detection. AEs are usually applied to urban, land cover, land use, forest, and building change detection. GANs are usually used for training data generation and can significantly improve the performance of deep learning networks, depending on the amount of training data. According to Table 2, the main limitations of previous deep learning methods include model complexity, time-consuming networks, poor performance in detecting the shape and boundary of objects, and small changes. Considering the limitations of previous studies, we propose a method based on semi-transfer learning. This method uses a pre-trained EfficientNetV2 T network [47] with faster training speed and efficient parameters as an encoder part for feature extraction and convolutional layers of Unet as a decoder part for the reconstruction binary change map. The Semi Transfer Learning Change Detection EfficientNetV2 T-Unet (STCD-EffV2T Unet) method not only has less complexity but also has a faster training speed. Besides, most previous studies have concentrated on urban and building change detection datasets with less complexity and diversity, while the area in our study was selected from northern Iran, in which the urban areas are highly complex. This area consists of three parts: the first part is the coastline and urban area near water or wetland, the second part is the urban area which is located on foothills and mountainous terrain, and the last part is the urban areas which are covered mainly by vegetation or are surrounded by forest and agricultural land. Our proposed method can detect changes with high precision in these complex urban areas. Furthermore, the performance of the proposed method is evaluated using OSCD, which is one of the most famous datasets for urban change detection.

The main purpose of this study is to detect changes in these three kinds of areas that have always been challenging, using our proposed method to overcome this challenge. In comparison with other studies, our proposed method is not time-consuming or complex. Moreover, even using a focal loss function that is not effective in terms of change detection results, its performance will be adequate. The advantages of using transfer learning with different inputs make our method stronger than the others.

The rest of this study is organized as follows. The Section 2 contains the datasets that are used in this research and the properties of the study areas. In the Section 3, our proposed method is investigated in detail and the results are demonstrated in the Section 4. The comparisons between the performances of the methods are collected in Section 5. The conclusion is included in the Section 6.

## 2. Materials and Datasets

### 2.1. Sentinel-2 Satellite Image of the Northern Iran Dataset

The Sentinel-2 satellite images from northern Iran's urban areas, especially located in the Gilan and Mazandaran provinces, are used in this study. The urban areas in these provinces are chiefly located on the margins of forests and farmlands as well as on the coastline, near water or wetland, and also on mountainous terrains or foothills. Most of the changes in these areas are located near forests and agricultural lands [48,49]. The changes are related to the expansion of urban areas horizontally due to the construction of tourism facilities, countryside houses, and resorts.

This research uses Sentinel-2 satellite images captured on 2 July 2017, and 10 August 2021, and their corresponding ground truth produced by using Google Earth images taken simultaneously. Change polygons are drawn employing ENVI and ArcGIS software. The Sentinel-2 images include 13 bands in the short-wave infrared, near-infrared, and ultraviolet parts of the spectrum, with resolutions ranging between 10 and 60 m (the multispectral satellite images are processed from level-0 to level-2A by Payload Data Ground Segment (PDGS). Only level-1C and level-2A products are released to users. In this study, we use the user available products as dataset). Figure 1 illustrates the study area. This dataset will be uploaded here: [http://rslab.ut.ac.ir, accessed on 20 February 2023].

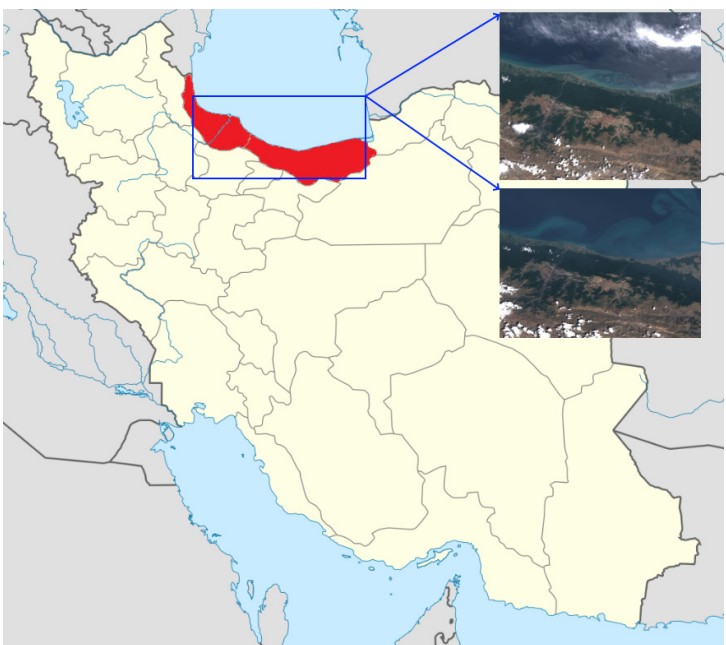

**Figure 1.** Map of study location in Iran (**red**), Sentinel-2 images of study location in 2017 (**top**) and 2021 (**bottom**).

### 2.2. Onera Sentinel-2 Satellite Change Detection (OSCD) Dataset

The second dataset adopted in this research is the OSCD dataset, which is a dataset of urban changes that can help to demonstrate the performance of our proposed method regarding complex urban changes. The OSCD dataset has been captured using Sentinel-2 images taken from places with various levels of urbanization in several other countries that have experienced rapid urban development and growth. This dataset includes the data of 24 important urban areas worldwide with ground truths. Figure 2 presents the RGB images of the five cities of Rio, Montpellier, Rennes, Chongqing, and Beihai at two times and their location in the world. The OSCD dataset is also available at https://rcdaudt.github.io/oscd/, accessed on 6 June 2020. In Table 3, the descriptions of the two datasets are illustrated.

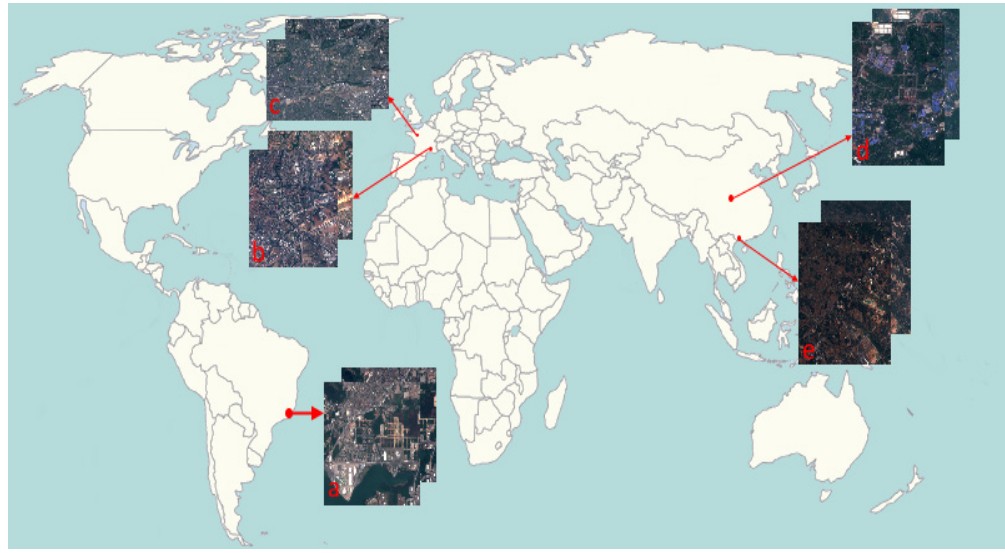

**Figure 2.** Multi-temporal image datasets. (**a**) Rio, (**b**) Montpellier, (**c**) Rennes, (**d**) Chongqing, and (**e**) Beihai.

**Table 3.** Information on datasets used in this study.

| Datasets | Time | | Bands | Spectrum Range (μm) | Spatial Resolution (m) |
|---|---|---|---|---|---|
| Onera Sentinel-2 satellite images Change Detection (OSCD) | Time1 | 2015 | Blue | 0.45~0.52 | 10 |
| | | | Green | 0.52~0.59 | |
| | Time2 | 2018 | Red | 0.63~0.69 | |
| North of Iran Sentinel-2 Satellite images | Time1 | 2017 | Blue | 0.45~0.52 | 10 |
| | | | Green | 0.52~0.59 | |
| | Time2 | 2021 | Red | 0.63~0.69 | |

## 3. Proposed Method

In this section, the proposed method is described in detail. As Figure 3 shows, the general procedure of the proposed method includes three main steps: (1) pre-processing, (2) data augmentation and train test split, and (3) encoder-decoder architecture. The main steps will be described in the following sub-sections.

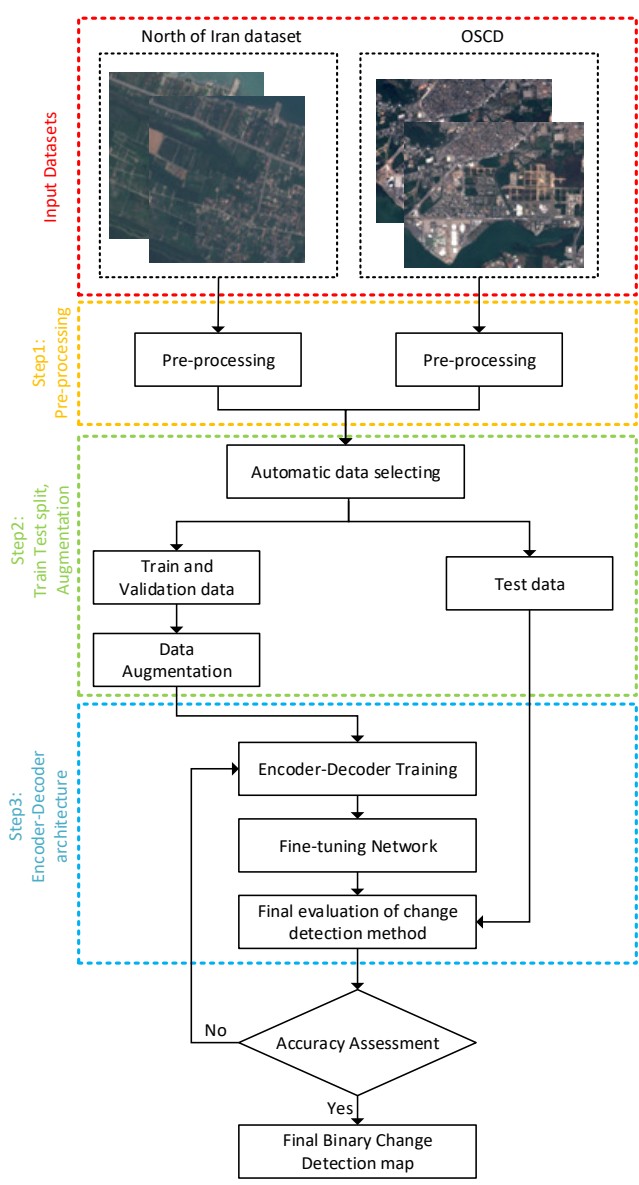

**Figure 3.** Flowchart of the proposed method.

### 3.1. Pre-Processing

In the pre-processing step, the RGB channels of two datasets are normalized using Equation (1), then they are divided into 128 × 128 patches, and, finally, the patches which contain the higher portion of the target in urban areas are chosen. Moreover, the ground truth is converted into 128 × 128 patches so that it is suitable as an input of the STCD-EffV2T Unet.

$$\frac{x - d_{min}}{d_{max} + d_{min}} \tag{1}$$

In Equation (1), $d_{min}$ is the minimum gray value in each channel, $d_{max}$ is the maximum gray value in each channel, and x is the gray value of each pixel.

### 3.2. Train and Test Split and Augmentation

After pre-processing, the evaluation and training data are separated from the test data. A total of 67% of all the data are considered as training and evaluation, and the remaining 33% are applied to testing. In the next step, augmentation is used for the training dataset, as a result of which the neural network will resist the changes in the augmentation domain and boost the network's efficiency [50]. Figure 4 shows data augmentation for a single image. The augmentation of this study includes rotation with three angles (+90°, −90°, and 180°).

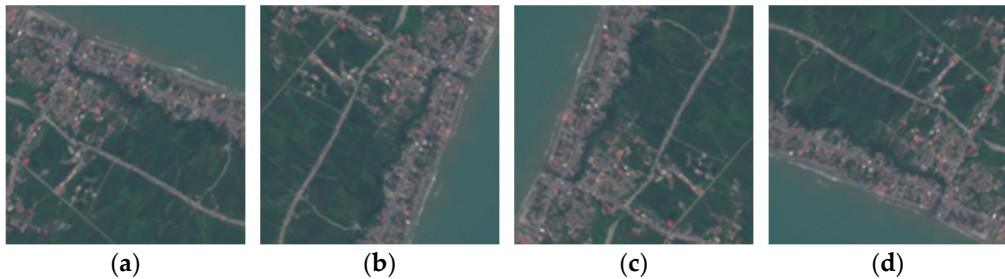

| (a) | (b) | (c) | (d) |

**Figure 4.** Original image (**a**); data augmentation with −90° (**b**), +90° (**c**), and 180° (**d**).

In the North of Iran dataset, the training patches raise from 1545 to 4140 and in OSCD they raise from 495 to 1980, in which surplus is the result of augmentation.

### 3.3. Encoder-Decoder Architecture

In this section, we describe the EfficientNetV2 T-Unet as an encoder-decoder architecture for binary change detection. EfficientNetV2 T is used as a feature extractor in the encoder stage and the convolutional layers of the Unet network are used as decoders.

#### 3.3.1. EfficientNet Encoder

The EfficientNet network families were proposed for the first time by [51]. In previous studies, the EfficientNetV1 as an encoder and convolutional layers of Unet as a decoder were used for semantic segmentation at environments [52], disease detection with CT scan and colonoscopy images [53,54], surface defect detection [55], blood vessel segmentation [56], and neuron instance segmentation [57]. In this study, we use the EfficientNetV2 as an encoder, which was proposed for the first time by [47]. In the EfficientNetV2, a combination of training-aware neural architecture reaches and scaling is used to optimize training speed and parameter efficiency. The EfficientNetV2 has versions B0, B1, B2, B3, L, M, S, and T. To obtain the maximum advantages of the EfficientNetV2 networks, we evaluate them as encoders of our proposed method (the results are presented in Section 4). Since EfficientNetV2 T comprises an efficient number of parameters and a high speed in the training process, it is chosen as the encoder to extract features. In addition, it reached better accuracy.

The fundamental building blocks of EfficientNetV2 are the mobile inverted bottleneck convolution (MBConv) [52] and fused mobile inverted bottleneck convolution (Fused-MBConv) [47]. The structures of these two blocks are shown in Figure 5.

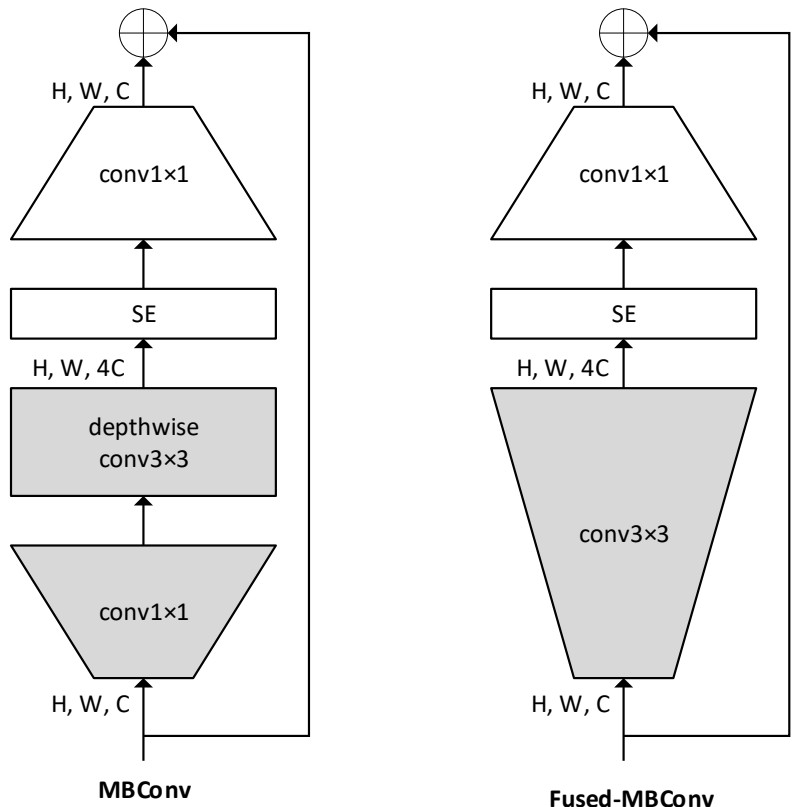

**Figure 5.** The structure of MBConv and Fused-MBConv blocks.

The EfficientNetV2 networks include seven blocks. In blocks 1–3, MBConv is replaced with Fused-MBConv; however, in the EfficientNetV1, all blocks are MBConv. Fused-MBConv improves training speed with a slight overhead on parameters and FLOPs. If, in all blocks (1–7), MBConv is replaced with Fused-MBConv, the training speed will reduce and increase the FLOPs and model complexity. The network architecture of the EfficientNetV2 T used in this study is illustrated in Figure 6.

### 3.3.2. Transfer Learning with Different Input Channels

Most models that are similar to EfficientNetV2 were trained by datasets such as ImageNet and have three channels as input. For change detection, the pre-trained model should be converted from three channels into six channels (three RGB Time1, three RGB Time2). In previous studies, the three methods that were used are as follows:

(1) Expand the weight dimensions to account for additional channels and randomly initialize the value.

(2) Similar to the first method; however, instead of using a random value for filling, using the mean of other values.

(3) A second parallel network is created which has the same architecture as the pre-trained network but has different input channels. This new parallel network performs feature extraction on the remaining channels. Then, the output of this network is concatenated with the output of the original pre-trained network. In this method, the parallel network learns the representation specific to additional channels, and we still take advantage of using the pre-trained model as well.

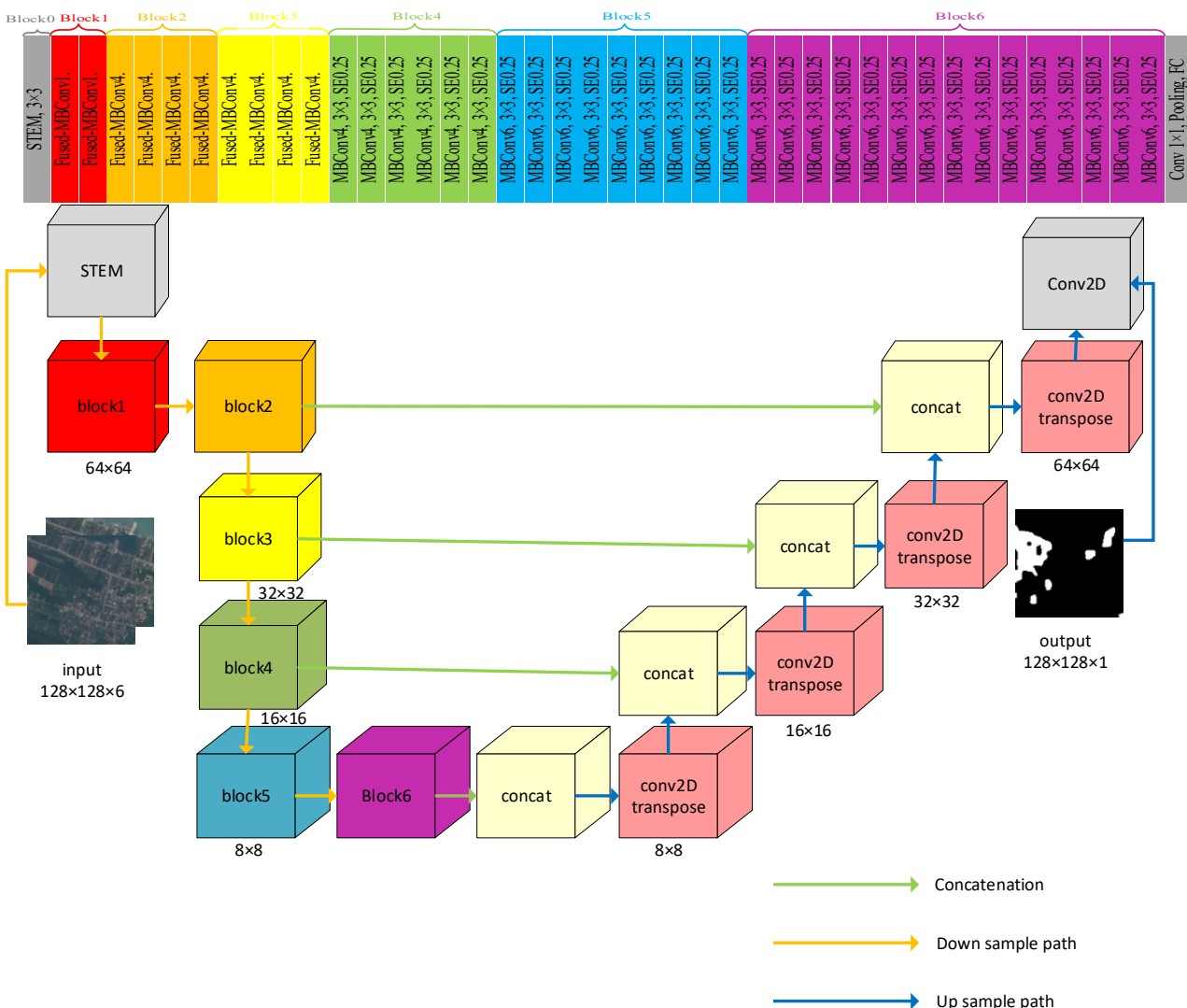

**Figure 6.** Top: the structure of EfficientNetV2 T, which contains seven blocks (represented with a different color); bottom: the structure of our proposed method using EfficientNetV2 T as an encoder and the convolutional layers of Unet as a decoder. The encoder is concatenated with the decoder at four different resolutions (Block2, Block3, Block4, and Block6).

In this study, we tried all three methods. For some networks, such as EfficientNet and Yolox, the second method has better performance; however, for DeepLabv3+ and VGG19, the third method works properly. In this study, we use the "Semi-Transfer Learning" technique in the sense that the encoder path (EfficientNet V2T) was pre-trained by the ImageNet dataset, although the decoder path (convolutional layers of Unet) was not trained.

### 3.3.3. Unet Decoder

Unet is a U-shaped, fully convolutional neural network which was used for the first time by Olaf and Ronneberger to segment medical images [58]. The Unet network includes the encoder and decoder paths. The task of the encoder path is to detect the backgrounds of images. The output of the encoder is smaller than the input; however, it is expanded in the second path, called the decoder. The second path detects the exact position of features by transpose convolution [59]. The decoder incorporates the high-level features by a subsequence of transpose and concatenation with the corresponding feature maps from the encoder path. The feature maps captured from the encoder path contain low-level features. In contrast, these low-level features contain helpful information about complex

scenes with multiple objects and their relative configuration. They are combined with intermediate high-level feature maps from Unet, the decoder path, and low-level feature maps from EfficientNetV2. The network boosts context information on higher resolution layers by using many feature channels in the up-sampling part [52].

Our proposed method uses EfficientNetV2 T as an encoder instead of a set of convolutional layers and the decoder part is the convolutional layers of Unet. Figure 6 illustrates the proposed architecture in detail.

### 3.3.4. Loss Function

We implement the focal binary loss function which is usually used for unbalanced binary classification [60]. The focal loss can significantly decrease the training and validation loss during the training process. This loss function is defined using the following equation:

$$\mathrm{FL}(\rho) = -\alpha(1 - \rho)^{\gamma} \log(\rho). \tag{2}$$

In Equation (2), $\rho \in [0, 1)$ is the model-estimated probability for the class with label 1 and $(1 - \rho)$ for the class with label 0, and $\alpha$ and $\gamma$ are two hyperparameters for which $\alpha$ shows a weight balancing factor for class 1 (default is 0.25 as mentioned in [60]) and $\gamma$ is a focusing parameter that is used to compute the focal factor (default is 2.0 as mentioned in [60]). In this study, we consider $\alpha = 0.1$ and $\gamma = 2.0$ since the performance of the loss function with $\alpha = 0.1$ is better than the default value. Moreover, the Adam optimizer with a learning rate of $1 \times 10^{-4}$ and decay rate of $1 \times 10^{-6}$ is considered to adjust network parameters.

### 3.3.5. Accuracy Assessment

Accuracy assessment is a non-separable part of any remote sensing task. In this study, the final results of the proposed method are compared with the ground truth and other change detection networks quantitatively and qualitatively. The quantitative comparison is based on the metrics which are described subsequently (Table 4).

**Table 4.** Information formulas for accuracy assessment metrics.

| Metric | Formula |
|---|---|
| Precision | $\frac{\mathrm{TP}}{\mathrm{TP+FP}}$ |
| F1-score | $\frac{2\times\mathrm{TP}}{(2\times\mathrm{TP})+\mathrm{FP}+\mathrm{FN}}$ |
| IOU | $\frac{\mathrm{TP}}{\mathrm{TP+FP+FN}}$ |
| Accuracy | $\frac{\mathrm{TP+TN}}{\mathrm{TP+FN+TN+FP}}$ |
| Kappa Coefficient (KC) | $\frac{2\times(\mathrm{TP}\times\mathrm{TN}-\mathrm{FN}\times\mathrm{FP})}{(\mathrm{TP+FP})\times(\mathrm{FP+TN})+(\mathrm{TP+FN})\times(\mathrm{FN+TN})}$ |

### 3.3.6. Comparative Methods

To compare the efficiency and speed of our proposed method, we compare our method with other EfficientNetV2 series including B0, B1, B2, B3, L, M, and S. Other EfficientNet series have a different number of parameters; however, they have a nearly similar general architecture. In addition, the following networks are used to confirm the efficiency of STCD-EffV2T Unet. These approaches are YoloX [61], ResNest [62], VGG19 [63], DeepLabv3+ [64], and U²Net [65], which are briefly described below:

- YoloX [61]: the Yolo family networks are generally used for object detection. These networks are fast and accurate, and trained on the COCO dataset. In this study, we use YoloX series such as Nano, Tiny, S, and X, which are trained on the COCO dataset as an encoder, and the convolutional layers of Unet as a decoder.
- ResNest [62]: this network, named the split-attention network, includes four series: ResNest50, ResNest101, ResNest200, and ResNest269. The number of parameters

increases according to the number of these networks. In this study, we use ResNest50, ResNest101, and ResNest200, which were trained by the ImageNet dataset as the encoder part and convolutional layers of Unet as the decoder part.

- VGG19 [63]: this network is one of the most famous networks for many remote sensing tasks. In this study, we use VGG19 which was pre-trained by ImageNet as an encoder path and convolutional layers of Unet as a decoder path.

- DeepLabV3+ [64]: the last modification of the DeepLab network is DeepLabV3+, uploaded at http://keras.io. This network is used for multiclass segmentation. In the architecture of the DeepLabV3+ network, the ResNet50 is the backbone which is pre-trained by ImageNet. In this study, we use DeepLabV3+ for binary change detection, and we change the input channels into six channels and share the weight with the third method in Section 3.3.4.

- $U^2$Net [65]: this network is one of the newest Unet network forms proposed for salient object detection. $U^2$Net is a two-level nested U-structure. In the structure of this network, different residual U blocks are used. We compare the performance of this approach with our proposed method.

## 4. Experimental Results

In this study, the STCD-EffV2T Unet is implemented using the RGB channels of two datasets. The system configuration used for this study is Intel (R) core (TM)i7-7800X CPU 3.5 GHz, 32.0 GB installed RAM, and NVIDIA GeForce GTX 1050Ti. All networks are trained using the TensorFlow 2.10.0 platform and python 3.8.

One of the main goals of this study is to develop a fast network. Our proposed method can generate an accurate binary change detection map in a reasonable time. The comparison between training time and several other parameters of this method with other methods is shown in Table 5. The metrics' results, introduced in Section 3.3.5, are also demonstrated.

**Table 5.** Quantitative evaluation of the results captured for two datasets.

| Method | Accuracy (%) | Precision (%) | F1-Score (%) | IOU (%) | Kappa Coefficient (KC) | Time of Training (h min s) | Parameters (Million) |
|---|---|---|---|---|---|---|---|
| STCD-EffV2T Unet (proposed method) North of Iran dataset | **97.66** | **99.61** | **98.79** | **97.60** | **0.67** | 2 h 10 min 34 s | 6.6 M |
| STCD-EffV2T Unet (proposed method) OSCD | 97.32 | 98.44 | 97.05 | 96.34 | 0.59 | 5 min 0 s | 6.6 M |
| EffV2 B0 Unet | 97.45 | 99.60 | 98.68 | 97.40 | 0.59 | 1 h 15 min 26 s | 4.3 M |
| EffV2 B1 Unet | 97.54 | 99.29 | 98.72 | 97.48 | 0.63 | 1 h 50 min 30 s | 4.8 M |
| EffV2 B2 Unet | 97.31 | 98.98 | 98.60 | 97.25 | 0.63 | 1 h 56 min 3 s | 5.2 M |
| EffV2 B3 Unet | 97.23 | 98.89 | 98.56 | 97.16 | 0.62 | 2 h 5 min 32 s | 6.2 M |
| EffV2 L Unet | 97.17 | 99.03 | 98.53 | 97.10 | 0.60 | 3 h 30 min 45 s | 26.0 M |
| EffV2 M Unet | 97.03 | 99.14 | 98.46 | 96.97 | 0.56 | 3 h 38 min 56 s | 13.7 M |
| EffV2 S Unet | 97.23 | 99.30 | 98.56 | 97.17 | 0.58 | 2 h 38 min 44 s | 8.8 M |
| YoloXNano Unet | 97.20 | 99.00 | 98.54 | 97.13 | 0.60 | 1 h 17 min 3 s | **2.1 M** |
| YoloXTiny Unet | 97.37 | 99.04 | 98.66 | 97.24 | 0.61 | **1 h 15 min 22 s** | 2.9 M |
| YoloXX Unet | 97.17 | 99.07 | 98.45 | 97.13 | 0.51 | 3 h 53 min 46 s | 20.4 M |
| YoloXS Unet | 97.38 | 99.08 | 98.65 | 97.34 | 0.55 | 1 h 18 min 41 s | 3.5 M |
| VGG19 Unet | 97.04 | 99.07 | 98.62 | 97.24 | 0.58 | 1 h 28 min 34 s | 18.2 M |

**Table 5.** *Cont.*

| Method | Accuracy (%) | Precision (%) | F1-Score (%) | IOU (%) | Kappa Coefficient (KC) | Time of Training (h min s) | Parameters (Million) |
|---|---|---|---|---|---|---|---|
| ResNest50 Unet | 97.05 | 99.05 | 98.47 | 97.15 | 0.62 | 3 h 21 min 26 s | 16.5 M |
| ResNest101 Unet | 97.27 | 99.09 | 98.58 | 97.21 | 0.61 | 3 h 41 min 0 s | 34.8 M |
| ResNest200 Unet | 95.57 | 97.15 | 97.67 | 95.46 | 0.50 | 9 h 36 min 34 s | 56.8 M |
| DeepLabV3+ | 92.82 | 95.03 | 96.21 | 92.70 | 0.29 | 2 h 30 min 2 s | 17.0 M |
| U$^2$Net | 97.37 | 99.03 | 98.60 | 97.30 | 0.60 | 6 h 2 min 18 s | 44.0 M |

Table 5 shows that the STCD-EffV2T Unet achieves an accuracy of 97.66% and KC achieves an accuracy of 0.67 after 100 epochs, which are the best results among other networks. Despite some exceptions, the training time increases by increasing the number of parameters. YoloXNano Unet, YoloXTiny Unet, and EffV2 B0 Unet have the shortest training time and their parameters are less than 5 M. However, the final change detection results show that these networks cannot overcome the complexity of the study areas. The precession, KC, and other metrics show that the performance of the STCD-EffV2T Unet is better than these three networks. In Table 5, we see that, when the number of the parameters exceeds eight million, the training of all the networks becomes time-consuming, whereas the final change detection map will not improve. The networks such as EffV2L Unet, EffV2M Unet, YoloXX Unet, U$^2$Net, and ResNest family networks are time-consuming in terms of their training process, and the parameters and complexity of these networks are surplus for our binary change detection task. Among the YoloX series, YoloXNano, YoloXTiny, and YoloXS, which have the lowest training time, cannot detect binary changes. YoloXX has more than 20 million parameters; however, its performance is not satisfactory. Among the EffV2 family networks, the STCD-EffV2T Unet compromises between the number of parameters and the binary change detection accuracy. The EffV2 B0, EffV2 B1, EffV2 B2, and EffV2 B3 have fewer parameters than STCD-EffV2T Unet, and, therefore, cannot reach an accurate binary change map, whereas EffV2 L, M, and S have more parameters than STCD-EffV2T Unet; therefore, they are time-consuming and cannot reach an accurate final change map. Moreover, U$^2$net and DeepLabV3+ are not as accurate as STCD-EffV2T Unet.

## 5. Discussion

The quantitative, qualitative, and visual analyses demonstrate that the STCD-EffV2T Unet works effectively in detecting changes in the urban areas of our datasets. The north of Iran dataset includes three parts: (1) coastal areas, which are near water and wetland, (2) urban areas which are covered with vegetation and surrounded by forest and agricultural land, and (3) urban areas which are located in foothills. In Figure 7, samples of these areas are considered in order to demonstrate the results. In Figure 7a,b, two coastal areas are considered. Detecting changes near water or the coast has always been challenging; however, our proposed method can dominate this challenge and detect these changes. Most of the changes in coastal areas are sparse and small; therefore, detecting them is difficult. However, our proposed method can detect them properly and preserve the edges almost correctly. Figure 7c–e shows the results of three urban area samples covered with vegetation or surrounded by forest or agricultural land. These kinds of areas cover most of the north of Iran; in fact, the majority of the changes took place in these areas. The changes in these urban areas are vast and most of them are the constructions of truism facilities and countryside houses on the border of urban areas; this trend has grown seriously in recent years. The changes in these areas are dense and our proposed method can detect most of them and properly preserve the edges. In Figure 7a,e (Time1) there are some clouds in images; however, our proposed method, without any pre-processing, can detect changes. In Figure 7f, the result of urban areas which are located on the foothills is shown. The urban areas on the foothills form a minority of the study areas and the changes in this area are

sparse. Most of this urban area is located in the mountain's shadow; therefore, dividing between shadow and changes is hard. However, our proposed method can defeat this challenge and detect the changes and preserve the edges properly.

We also show the performance of STCD-EffV2T Unet on OSCD. The results of our proposed method for five cities of OSCD are shown in Figure 8. Although this network is slightly ill-suited to the detection of continued changes and those in edge or border areas (e.g., continuous changes in Montpellier City), it has considerable accuracy in tracking changes. As demonstrated by Figure 8, the network has effectively detected many tiny changes in Beihai and Rennes, where there have been many tiny changes in the ground truth. As shown in Figure 8, the proposed method also performed relatively well in detecting changes in coastal and port cities such as Rio.

Figure 9 compares the performance of STCD-EffV2T Unet with other EffV2 networks. It shows five coastal ports (a–e), five urban areas covered or surrounded by forest or agricultural land (f–j), and two urban areas located on foothills (k and l). These 12 areas are in the north of Iran. Among the EffV2 series, EffV2B3 and EffV2B2 can detect changes nearly properly; however, the STCD-EffV2T Unet can detect changes more accurately. As shown by Figure 9, the STCD-EffV2T Unet can preserve the edges and shapes of changes. There is the highest similarity between STCD-EffV2T Unet results and the ground truth.

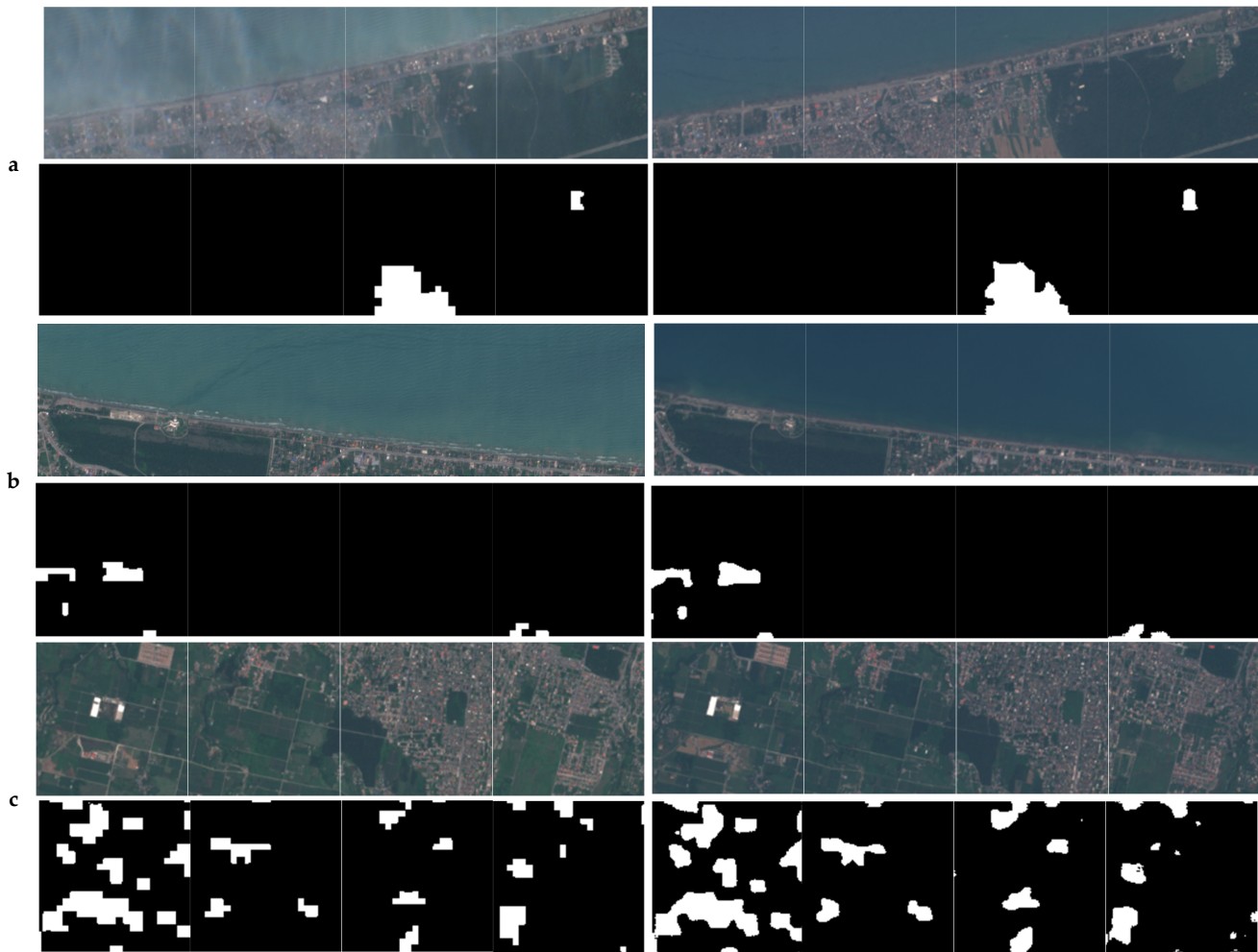

**Figure 7.** *Cont.*

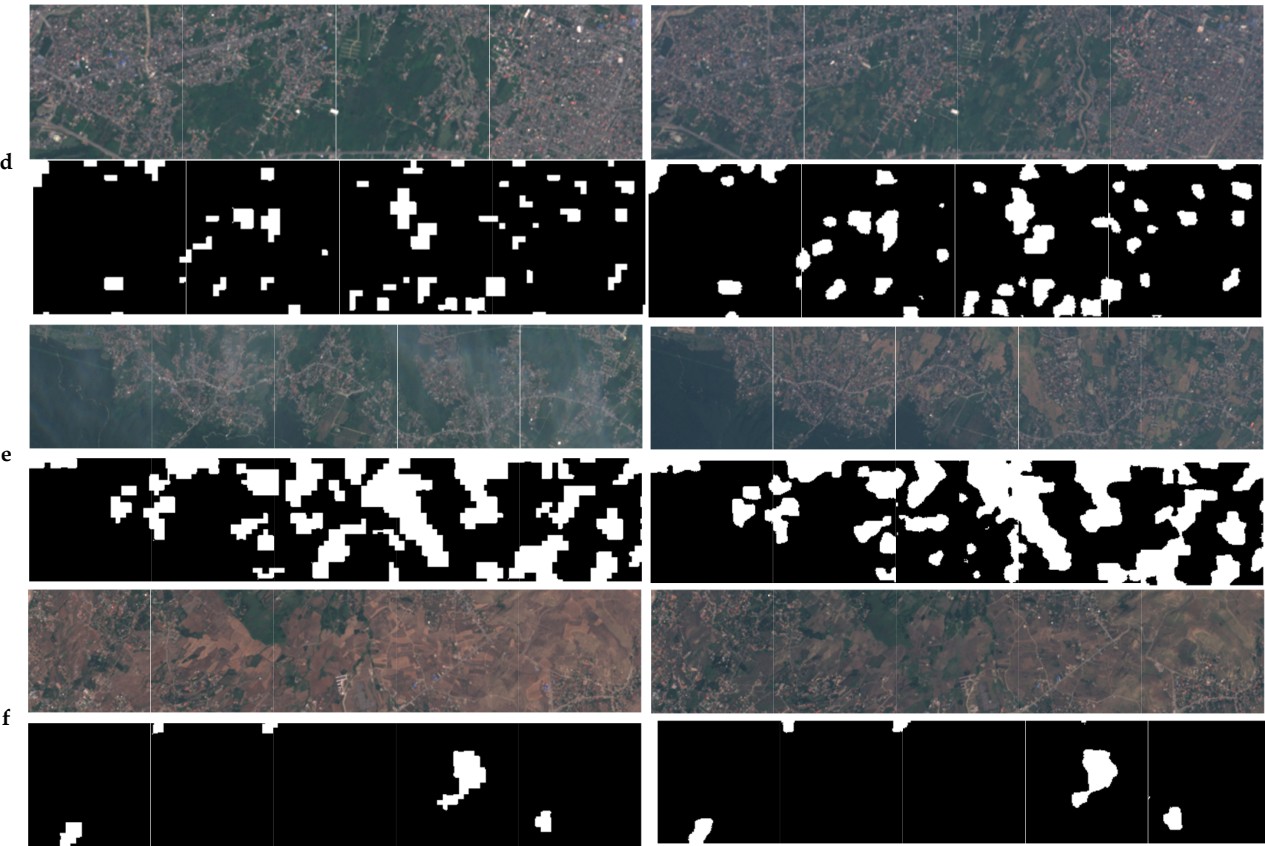

**Figure 7.** Proposed method results for coastal areas (**a**,**b**), urban areas surrounded by forest or agricultural land (**c**–**e**) and urban areas located on foothills (**f**) (top row contains Time 1 (2017) and Time 2 (2021) images, respectively; bottom row contains ground truth and proposed method results, respectively).

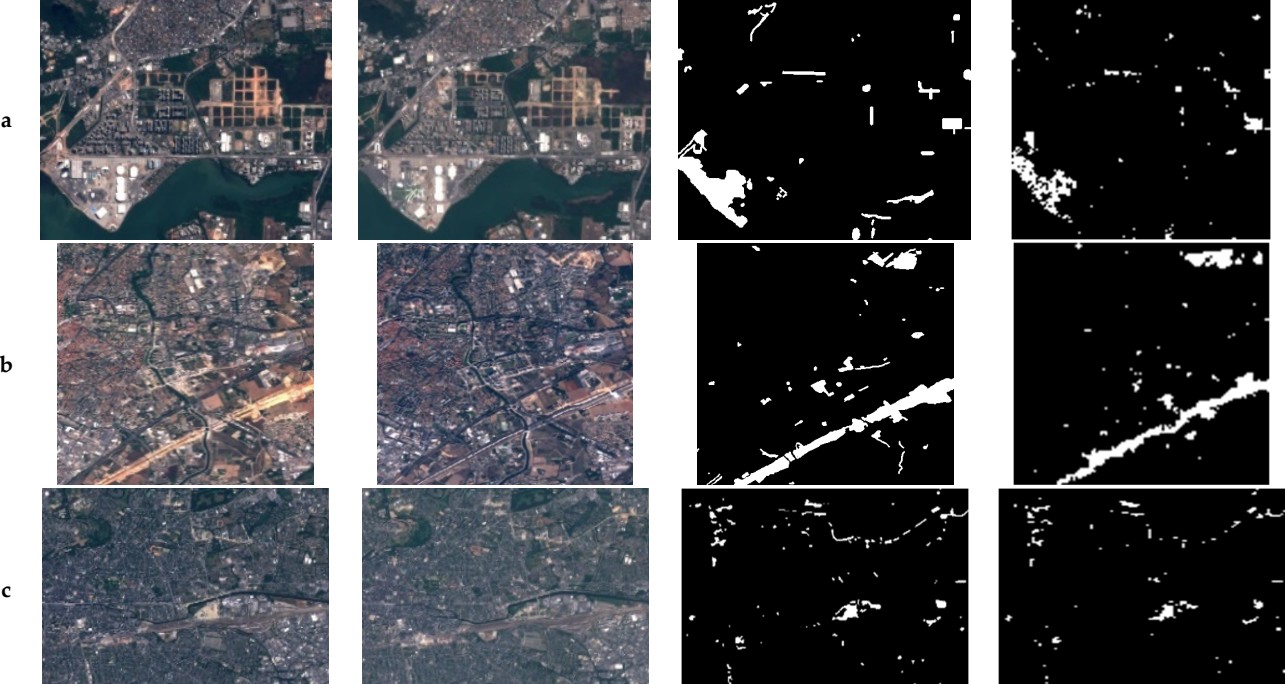

**Figure 8.** *Cont.*

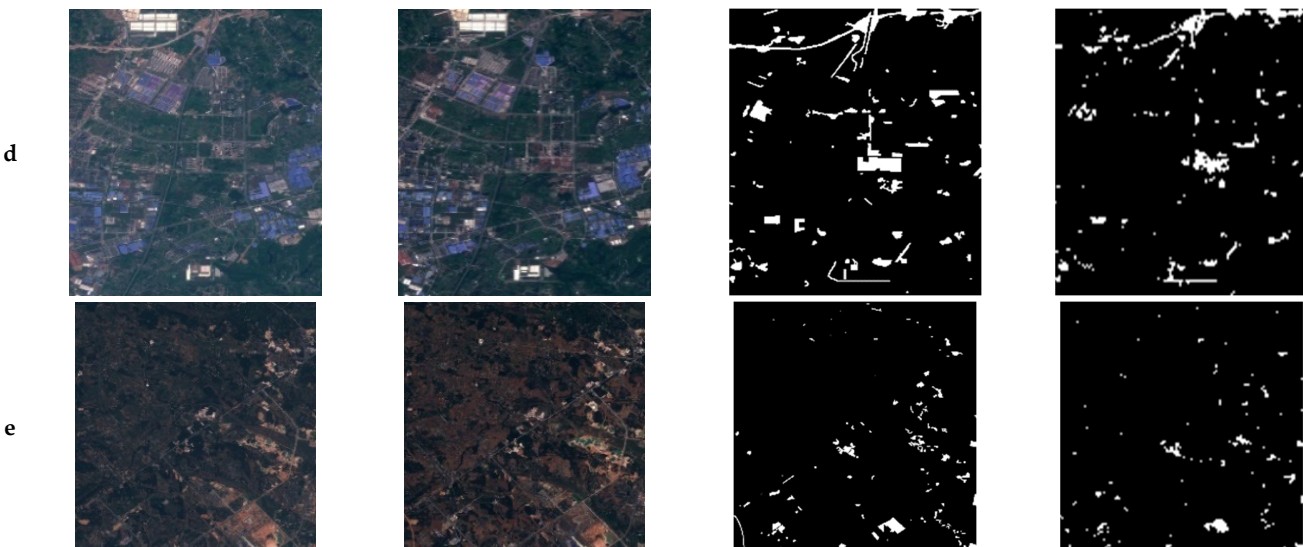

**Figure 8.** Results of the proposed method in the cities of Rio (**a**), Montpellier (**b**), Rennes (**c**), Chongqing (**d**), and Beihai (**e**) (from left to right: Time 1 image, Time 2 image, ground truth, and proposed method results).

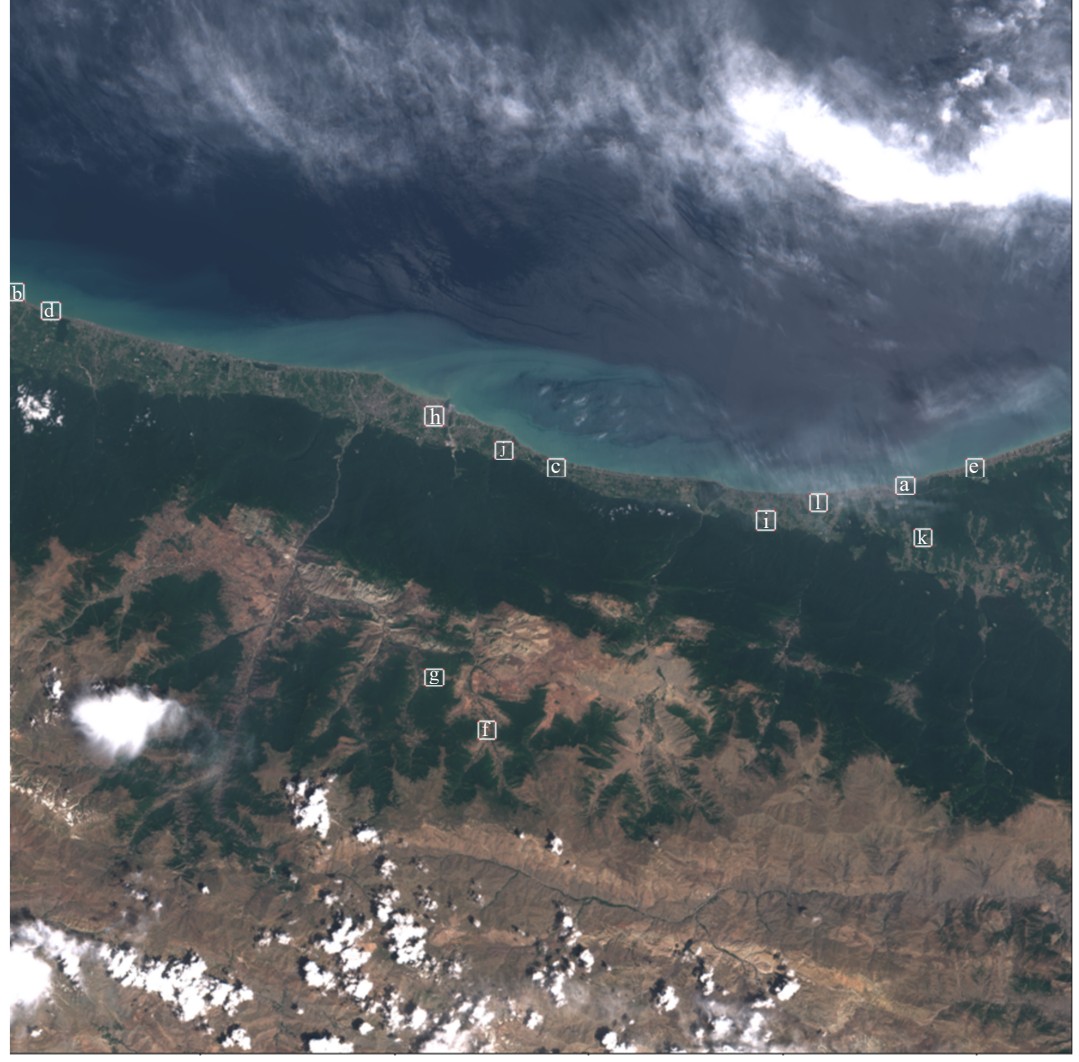

**Figure 9.** *Cont.*

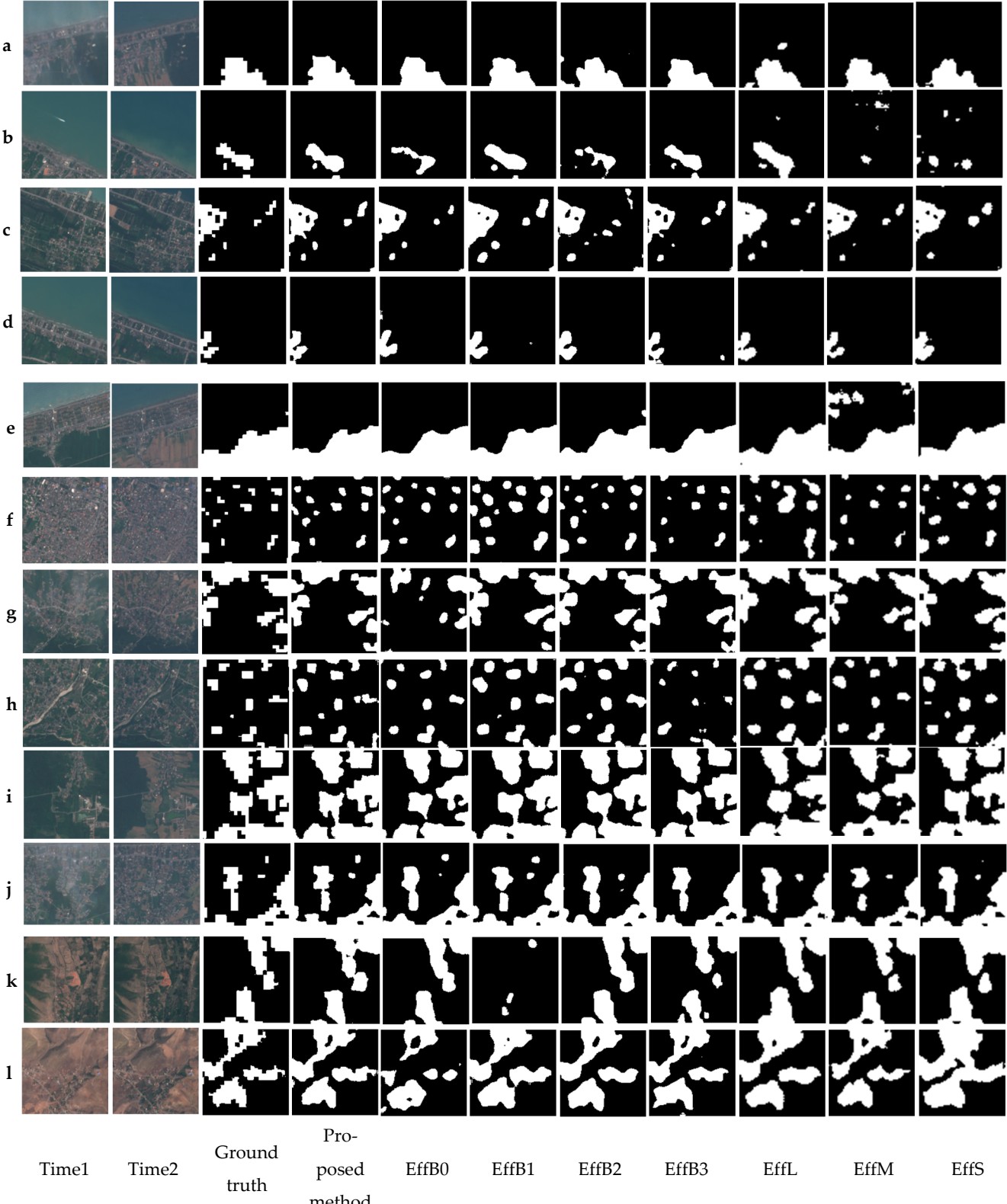

**Figure 9.** Results of coastal areas (**a**–**e**), results of areas located near the forest and agricultural land (**f**–**j**), and results of areas which are located on the foothills (**k**,**l**).

Figure 10 compares the performance of STCD-EffV2T Unet with the other networks, including the YoloX series, ResNest series, VGG19, U$^2$Net, and DeepLabV3+. In Figure 10, the same study areas are selected which were used in Figure 9. YoloXNano and YoloXTiny

have better performance among other methods; however, the STCD-EffV2T Unet is superior. The other methods are not only time-consuming, as reported in Section 4 (experimental results), but also cannot reach the best performance.

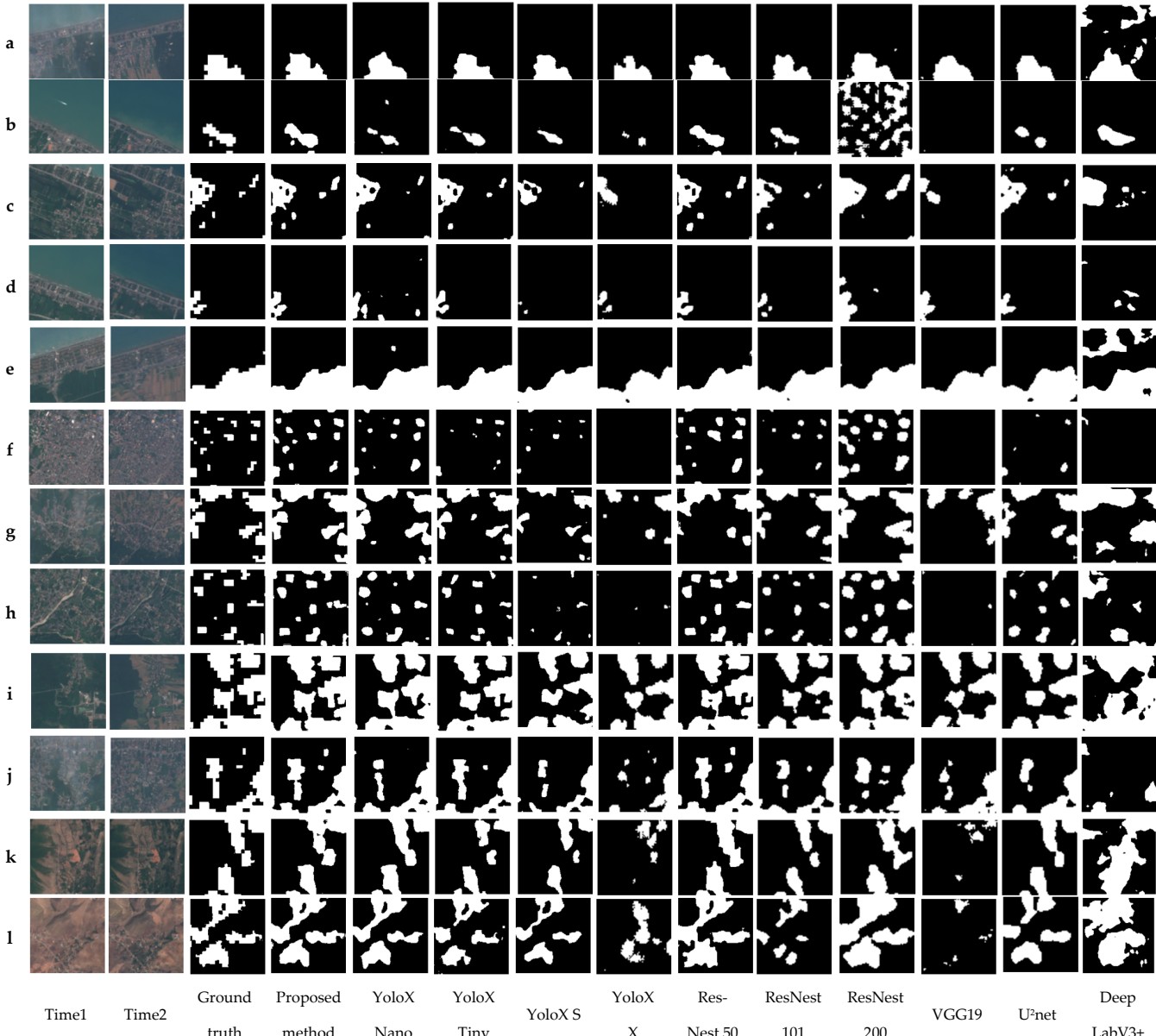

**Figure 10.** Results of coastal areas (**a–e**), results of areas located near the forest and agricultural land (**f–j**), and results of areas which are located on the foothills (**k,l**).

## 6. Conclusions

In this study, we developed the STCD-EffV2T Unet network architecture, which benefits from the advantages of transfer learning for urban change detection with the northern area of Iran and the OSCD dataset. In our method, the RGB channels of two datasets were assigned as input of the STCD-EffV2T Unet network. The STCD-EffV2T Unet network can detect changes effectively in complex urban areas by unique feature extraction through the EfficientNetV2, pre-trained by the ImageNet dataset in the encoding path and convolutional layers of Unet as a decoding path. Taking into account the training speed of the STCD-EffV2T Unet network, it can produce an accurate binary change map in a reasonable time. It also can serve as a valuable network for generating maps of urban northern areas of Iran and updating the RS and GIS databases. In addition to the speed

improvement, it achieves a 97.66% and 98.79% accuracy and F1-score, respectively. The STCD-EffV2T Unet successfully detected the diversity changes in northern Iran's urban areas, including port cities and the coastal regions, urban areas adjacent to the forests and agricultural lands, and urban areas located on foothills. In this study, we applied the focal loss function, which is particularly useful for simultaneously detecting small and dense changes and also preserving the edges. In future studies, researchers can develop and use STCD-EffV2T Unet to analyze other urban areas with different challenges or three-dimensional changes. It may also be worthwhile to design other deep learning method architectures, such as encoder-decoder networks, for binary change detection.

**Author Contributions:** All authors contributed to the study's conception and design. M.G., M.H. and P.R. performed material preparation, data collection, and analysis. M.G. wrote the first draft of the manuscript. All authors have read and agreed to the published version of the manuscript.

**Funding:** This research received no external funding.

**Data Availability Statement:** The datasets generated and analyzed during the current study are available from the corresponding author upon reasonable request.

**Acknowledgments:** The author would like to thank the Editor and reviewers for their valuable comments on our manuscript.

**Conflicts of Interest:** The authors declare no conflict of interest.

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
