# Peer review of "STCD-EffV2T Unet: Semi Transfer Learning EfficientNetV2 T-Unet Network for Urban/Land Cover Change Detection Using Sentinel-2 Satellite Images"

_remotesensing, doi:10.3390/rs15051232_

Round 1

Reviewer 1 Report

1. What is the main question addressed by the research?

The use of RS technology to track changes in land cover especially in urban Land cover is the main point of article. In the title “urban change detection” better to explain “urban built area” or “urban land cover Change(LC)”.

2. Do you consider the topic original or relevant in the field? Does it address a specific gap in the field?

Many research had done on LU/LC in GIS and RS. One of the main limitation of article is how use of TensorFlow 2.10.0 platform. It’s developed in Python by authors or they had used a developed tool?  

3. What specific improvements should the authors consider regarding the methodology? What further controls should be considered?

Change the title of work based on Q.1.

Add specific link in the Data Availability Statement to tool if it is available for all.

4. Are the conclusions consistent with the evidence and arguments presented and do they address the main question posed?

Conclusions are consistent with the evidence and arguments presented.

5. Are the references appropriate?

References are appropriate.

Author Response

Response to Reviewer 1:

  1. What is the main question addressed by the research?

The use of RS technology to track changes in land cover especially in urban Land cover is the main point of article. In the title “urban change detection” better to explain “urban built area” or “urban land cover Change (LC)”.

Answer 1: Thank you for the suggestion. In this paper, we focus on both urban and land cover change detection. According to your suggestion, we changed it to “STCD-EffV2T Unet: Semi Transfer Learning EfficientNetV2 T-Unet Network for Urban/Land Cover Change Detection using Sentinel-2 Satellite Images”.

  1. Do you consider the topic original or relevant in the field? Does it address a specific gap in the field?

Many research had done on LU/LC in GIS and RS. One of the main limitation of article is how use of TensorFlow 2.10.0 platform. It’s developed in Python by authors or they had used a developed tool?  

Answer 2: Thank you for pointing this out. We developed the networks which were pre-trained and available in TensorFlow 2.10.0 (TensorFlow 2.10.0 works in Python 3.8); i.e., its networks are developed according to the input channels, as well as sharing the weights.

  1. What specific improvements should the authors consider regarding the methodology? What further controls should be considered?

Change the title of work based on Q.1.

Add specific link in the Data Availability Statement to tool if it is available for all.

Answer 3: I appreciate your comment. The title was changed, and all datasets which were used in this study are available at: http://rslab.ut.ac.ir and https://rcdaudt.github.io/oscd.

  1. Are the conclusions consistent with the evidence and arguments presented and do they address the main question posed?

Conclusions are consistent with the evidence and arguments presented.

  1. Are the references appropriate?

References are appropriate.

Reviewer 2 Report

Dear authors, 

The article deals with an important issue in urban planning, change detection based on remote sensing data. The authors propose a methodology for change detection that combines the methodology of EfficientNetV2 with Unet. The working methodology is clearly defined, but I would have a few comments:
- The introduction contains an extensive synthetic literature review, and various change detection methodologies are presented in tables, but there is very little explanation in the text itself. -
I recommend that the article be accepted after a minor revision
Kind regards

Author Response

Response to Reviewer 2:

The article deals with an important issue in urban planning, change detection based on remote sensing data. The authors propose a methodology for change detection that combines the methodology of EfficientNetV2 with Unet. The working methodology is clearly defined, but I would have a few comments:
- The introduction contains an extensive synthetic literature review, and various change detection methodologies are presented in tables, but there is very little explanation in the text itself.

- I recommend that the article be accepted after a minor revision

Kind regards

Answer 1: Thank you very much for your comments. A summary explanation of all methods that were mentioned in Table 2 was added on page 6 (as highlighted). We also try to categorize all state-of-the-art methods in Tables 1 and 2 so easier to access and compare.

Reviewer 3 Report

·        You must include more information for the Sentinel-2 images you used such as acquisition date, product level of Sentinel, etc.

·        It would be informative to the reader to refer to what augmentation offers as a procedure and include more references (now you have only one).

·        I suggest to check Figure 6. In the legend you say that the structure contains 7 blocks. I see 6 colors and in the top all colors refer to Block 2.

·        Sentinel images in Figure 7 seem to have distortions and they are not clear, making difficult for the reader to understand the areas you describe.

·        It would be better to enlarge images (OSCD data, ground truth and results) in Figure 8, so that it will be possible for the reader to understand the landscape and particularly the urban areas.

·        In several parts of your paper you write that you used Unet decoder. I  suggest to think about using instead “…convolutional layers of Unet as decoding path.”, that you include in your Conclusion.

·        In Figure 9 the red squares with letters in them (I suppose) are not readable. The images a-l are too small. Perhaps an enlargement for one area of each category could be informative for the reader.

·        Check all the legends.

·        You need to examine very carefully your text and polish the language. There are spelling mistakes (for example grand truth instead of ground truth) and bad syntax (very long sentences, missing of verbs, wrong use of commas, etc).

Author Response

Response to Reviewer 3:

  1. You must include more information for the Sentinel-2 images you used such as acquisition date, product level of Sentinel, etc.

Answer 1: Thank you for pointing this out. The exact time of datasets acquisition is July 2, 2017, and August 10, 2021, which were mentioned in the revised manuscript. Also, the multispectral satellite images are processed from level-0 to level-2A by Payload Data Ground Segment (PDGS). Only level-1C and level-2A products are released to users. In this study, we use the user-available products as a dataset and this explanation is also added in the footnote of the revised manuscript file.

  1. It would be informative to the reader to refer to what augmentation offers as a procedure and include more references (now you have only one).

Answer 2: Thank you for this suggestion. In the case of our study, we used only rotation with three angles as the augmentation, which was mentioned in the manuscript.

  1. I suggest to check Figure 6. In the legend you say that the structure contains 7 blocks. I see 6 colors and in the top all colors refer to Block 2.

Answer 3: Thank you for pointing this out. The name “block0” (first gray block or STEM) was added and pointed out in Figure 6.

  1. Sentinel images in Figure 7 seem to have distortions and they are not clear, making difficult for the reader to understand the areas you describe.

Answer 4: By regarding your valuable comment, Figure 7 was totally changed. We show the blocks in a better way and quality.

  1. It would be better to enlarge the images (OSCD data, ground truth and results) in Figure 8, so that it will be possible for the reader to understand the landscape and particularly the urban areas.

Answer 5: Thank you for this suggestion. OSCD data and results were enlarged.

  1. In several parts of your paper you write that you used Unet decoder. I suggest to think about using instead “…convolutional layers of Unet as decoding path.”, that you include in your Conclusion.

Answer 6: Thank you for your helpful suggestion. We applied your suggestion everywhere it was relevant.

  1. In Figure 9 the red squares with letters in them (I suppose) are not readable. The images a-l are too small. Perhaps an enlargement for one area of each category could be informative for the reader.

Answer 7: Thank you for pointing this out. We changed the color of the blocks so that they could be clear and readable in contrast with the background image.

  1. Check all the legends.

You need to examine very carefully your text and polish the language. There are spelling mistakes (for example grand truth instead of ground truth) and bad syntax (very long sentences, missing of verbs, wrong use of commas, etc).

Answer 8: Thank you for pointing this out. We tried to improve the manuscript for proper English language, grammar, punctuation, spelling, and overall style. We appreciated any better suggestions as well.